



# Insights into new particle formation in Siberian boreal forest from nanoparticle ranking analysis

Anastasia Lampilahti[1], Olga Garmash[2], Diego Aliaga[1], Mikhail Arshinov[3], Denis Davydov[3], Boris Belan[3], Janne Lampilahti[1], Veli-Matti Kerminen[1], Tuukka Petäjä[1], Markku Kulmala[1,4,5] and Ekaterina Ezhova[1]

[1]Institute for Atmospheric and Earth System Research (INAR/Physics), University of Helsinki, Helsinki, P.O. Box 64, FI-00014, Finland

[2]Department of Chemistry, University of Copenhagen, Copenhagen, DK-2100, Denmark

[3]V.E. Zuev Institute of Atmospheric Optics of Siberian Branch of the Russian Academy of Science (IAO SB RAS), Tomsk, 634055, Russia

[4]Beijing University of Chemical Technology, Beijing, 100029, China

[5]Nanjing University, Nanjing, 210023, China

*Correspondence to*: Anastasia Lampilahti (anastasiia.lampilahti@helsinki.fi)

**Abstract.** New particle formation (NPF) plays a critical role in atmospheric processes and climate dynamics. Its mechanisms and impacts remain poorly understood in remote regions like Siberia. In this study, we used the data set from a long-term campaign (2019-2021) employing particle spectrometers (NAIS and DMPS) to investigate NPF at a boreal forest site in Western Siberia. So far, this is the longest dataset for statistics of Siberian NPF. We classified NPF events, calculated formation and growth rates, and performed nanoparticle ranking analysis. Similar to other boreal sites, spring is the most favorable period for NPF events in Siberia. We observed a seasonal variability in growth rates, with the higher values in summer and the lower values in winter. We showed that the results of the ranking analysis can be used to identify the days with high or low NPF event probability, similar to the previous results obtained on the data set from the Finnish boreal forest (SMEAR II station). Nanoparticle ranking analysis introduces aa new metric, $\Delta N_{2.5-5}$, which is the daily maximum concentration of particles in 2.5–5 nm range with subtracted background concentration and is linked with both probability and intensity of NPF. In order to identify the factors influencing NPF in Siberia, we analyzed the correlation between $\Delta N_{2.5-5}$ and concentrations of trace gases, such as $SO_2$, $O_3$, $NO$, $NO_2$, as well as global solar radiation, temperature, relative humidity (RH), and wind speed. We investigated the dependence of particle formation rate ($J_3$) on $\Delta N_{2.5–5}$, finding a strong positive correlation confirmingconfirming the connection of $\Delta N_{2.5–5}$ with the probability and intensity of NPF. $SO_2$, linked to anthropogenic pollution, played a significant role in spring when most of NPF events wewere observed. Ozone correlated positively with $\Delta N_{2.5–5}$ in spring and summer, likely due to VOC oxidation. NOx showed seasonally variable effects, with NO positively influencing NPF in autumn and $NO_2$ showing both positive and negative correlations depending on the season. Global solar radiation significantly enhanced NPF by driving photochemical reactions leading toto sulfuric acid production. Temperature suppressed NPF in spring and summer, aligning with the SMEAR II findings. RH had a negative influence across seasons, while condensation sink suppressed NPF, particularly in winter when its values peaked. Sulfuric acid calculated via proxy, critical for nucleation and growth, was a key driver of NPF in winter, spring, and autumn. These findings provide a comprehensive understanding of



NPF processes in Siberia and highlight the importance of long-term datasets for uncovering regional and seasonal patterns in aerosol formation and growth.

## 1. Introduction

New Particle Formation (NPF) is a phenomenon in which new aerosol particles are formed due to the gas-to-particle conversion influencing atmospheric aerosol particle population (Kulmala et al., 2014). Aerosols can scatter solar radiation, but some of the aerosols can absorb solar radiation (Myhre et al., 2013). Aerosols that mainly scatter solar radiation have a cooling effect on climate (IPCC 2021). Aerosols also have impact on clouds, because they can act
as cloud condensation nuclei (CCN) (Merikanto et al., 2009, Kazil et al., 2010, Kerminen et al., 2012), and have a significant influence on Earth radiation budget and climate (Makkonen et al., 2012, Dunne et al., 2016, Gordon et al., 2017). NPF occurs in different environments (Kerminen et al., 2018); one of the well-studied environments is boreal forest, because NPF is often associated with biogenic emissions of volatile organic compounds (Bäck et al. 2012, Tunved et al., 2006, Mäki et al., 2019). A significant  of the global boreal forests are located in Siberia, Russia;
however, our knowledge is largely based onmeasurements conducted at thethe European sites, such as SMEAR II station in Hyytiälä, Finland (Hari and Kulmala, 2005), or SMEAR Estonia in Järvselja, Estonia (Noe et al., 2015).

For investigating NPF processes, the classification method described byby Dal Maso et al. (2005) is common and the guidelines for using this method are described in Kulmala et al. (2012). For calculating NPF event frequency, all the days when the measurements are conducted are usually divided into three categories: NPF event days, when
formation and growth are clearly observed; non-event days, when no formation or growth happens; and undefined days. Undefined days are those that contain other types of events like "tail", "apple", or "bump" (Buenrostro Mazon et al., 2009, Yli-Juuti et al., 2009). We refer to this classification as "traditional", because it is widely used in the literature (Vana et al., 2016, Dada et al., 2017, Cai et al., 2017, Kerminen et al., 2018, Nieminen et al., 2018, Deng et al., 2020, Bousiotis et al., 2021). The typical annual NPF event frequencies in boreal forest regions vary from
10% to 30 % (Kerminen et al., 2018, Artaxo et al., 2022). The yearly average NPF event frequency at SMEAR II station is 26% (Dal Maso et al., 2005, Vana et al., 2016, Nieminen et al., 2018), and at SMEAR Estonia, it is about 21% (Vana et al., 2016).

Aerosol-related studies in Siberia werewere mostly performed using the data from Zotino Tall Tower Observatory (ZOTTO) (Heintzenberg et al., 2011, Chi et al., 2013, Mikhailov et al., 2015, Wiedensohler et al., 2019) and
Fonovaya station (Buchelnikov et al., 2020, Arshinov et al., 2021, Arshinov et al., 2022, Lampilahti et al., 2023, Garmash et al., 2024). Wiedensohler et al. (2019) reported very low annual NPF event frequencies at ZOTTO, only 3% of days were classified as events. OurOur previous study at Fonovaya station showed that NPF on average occurs in less than 10% of days (Lampilahti et al., 2023). WeWe showed that high values of sky clearness index and high concentrations of trace gases, especially $SO_2$, $NO_2$, and NO, have the largest impact on Siberian NPF in spring. Also,
important NPF properties such as growth rates (GR) and formation rates (J) at 5 to 20 nm particle diameter were reported. However, GR and J in Lampilahti et al. (2023) were calculated using the data from the Diffusional Particle Sizer (DPS). This instrument measures particle number size distribution from 3 nm to 200 nm with 20 size bins, and



its resolution is not enough for rigorous calculations. Because of this, the GR and J values, calculated using the data from this instrument, might be less accurate than those calculated from Neutral cluster Air Ion Spectrometer (NAIS),

what measures particle and ion size distributions from 2 nm to 40 nm with 24 size bins (Carracedo et al., 2022) using appearance time method (Lehtipalo et al., 2014).

Our recent study based on the data from Fonovaya station showed unexpectedly high monthly NPF frequency (50% of days in March werewere event daysdays) during early spring caused by the Siberian heatwave in 2020 (Garmash et al., 2024). That study showed thatthat vapors, such as sulfuric acid, ammonia, biogenic organic vapors, contribute

to the particle formation at at this site. The warmer temperatures during the spring heatwave triggered biogenic activity that enhanced NPF event frequency in air masses from polluted areas. Interestingly, frequent NPF in Siberia occurred in polluted masses, whereas at SMEAR II station in the Finnish boreal forest, NPF occurs in the air masses from the clean sector (Vana et al., 2016).

Most of the previous studies focusing on NPF in different environments have used the traditional NPF classification

method discussed above. It has certain disadvantages: classification is done manually, that can bring human bias to the results. In this study, alongside with the traditional classification, we also used nanoparticle ranking method. Nanoparticle ranking, introduced by Aliaga et al. (2023), based on the data from the Finnish station SMEAR II, uses the variable $\Delta N_{2.5\text{-}5}$, calculated from the particle number concentration at sizes from 2.5 to 5 nm, which is shown to be tightly linked to the occurrence probability and intensity of atmospheric NPF events. Nanoparticle ranking

method is objective, quantifiable and replicable, and it provides a representative value for each measurement day. Another advantage of nanoparticle ranking is that the days are not divided into 3 categories like in the traditional classification but rather represented in a probabilistic framework. This method provides a continuous variable where at one side, most of the days can be classified as non-events, and at another – as events.

In this study, we use a new data set from the measurement campaign at Fonovaya station spanning 2 years (July

2019 – November 2021) to get a better insight into NPF taking place in Siberia. We analyze NPF statistics using two methods and determine particle formation (J) and growth rates (GR) using more precise calculations, utilizing data from high-resolution instruments, providing better accuracy compared to our previous study (Lampilahti et al., 2023). Furthermore, we explore the seasonal differences in NPF events and use ranking method to analyze the link of various atmospheric parameters to $\Delta N_{2.5\text{-}5}$ representing the occurrence of NPF events and identify atmospheric

conditions that favor NPF in Western Siberia during different seasons.

## 2. Materials and methods

### 2.1 Observation sites

In the current study we used data collected at Fonovaya station in West Siberia, Russia. The station (56°25"N, 84°04"E) is located in Tomsk region, Russia. The description of the station can be found in Antonovich et al. (2018)

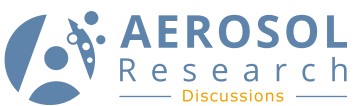

and Lampilahti et al. (2023). The closest cities are Tomsk (60 km East from the station, about 600 000 inhabitants), and Novosibirsk (170 km South - South - West, 1 200 000 inhabitants). The measurement site is situated on theeast bank of the Ob river and surrounded by the mixed boreal forest.

**2.2 Instrumentation**

In 2019-2021, INAR and IAO undertook a measurement campaign at Fonovaya station to perform more accurate and
comprehensive analysis. The following instrument suite was used: Neutral cluster Air Ion Spectrometer (NAIS), Particle Size Magnifier (PSM), Differential Mobility Particle Sizer (DMPS), and Chemical zIonization Atmospheric Pressure interface Time-Of-Flight Mass Spectrometer (CI-APi-TOF). PSM and DMPS allowed particle number size distributions to be measured in a wider size range. Here, we present for the first time the analysis of the two-year detailed dataset of aerosol measurements using NAIS and DMPS.

**Table 1. Variables and corresponding instrumentation used in this study**

| Parameters | Instrument | Reference |
|---|---|---|
| particle number size distribution (sizes 3 nm – 200 nm) | DPS, (Diffusional Particle Sizer=Diffusion Battery + CPC) | Reischl et al., 1991<br><br>Ankilov et al., 2002 |
| particle number size distribution (sizes 2 nm – 40 nm) and ion number size distribution (mobility range 3.2–0.001 $cm^2$ $V^{-1}$ $s^{-1}$) | NAIS (Neutral cluster Air Ion Spectrometer) | Manninen et al., 2009 Mirme and Mirme, 2011 |
| particle number size distribution (sizes 7 nm – 1 $\mu$m) | DMPS (Differential Mobility Particle Sizer) | Aalto et al., 2001 |
| particle number size distribution (sizes 300 nm – 20 $\mu$m) | OPC (Grimm Aerosol Spectrometer Model 1.108, Optical Particle Counter) | |
| global solar radiation | Kipp and Zonen CM3 pyranometer | |



| air temperature and relative humidity | Vaisala HMP155 | |
|---|---|---|
| wind velocity | Young Model 85004 | |
| $O_3$ concentration | Optec 3.02 P-A | |
| $NO_x$ concentration | Thermo Scientific Model 42i-TL | |
| $SO_2$ concentration | Thermo Scientific Model 43i-TLE | |

The instruments we used in the present study are listed in Table 1. For measuring particle and ion size distributions we used Neutral cluster Air Ion Spectrometer (NAIS, Airel OÜ) (Manninen et al., 2009, Mirme and Mirme, 2011). NAIS measures number size distribution of aerosol particles withinwithin a size range from 2.0 to 40 nm, and also number size distribution of positive and negative ions with the electric mobility range withinwithin 3.2–0.001 $cm^2$

$V^{-1} s^{-1}$, corresponding to 0.8–40 nm (Millikan-Fuchs equivalent diameter, Mäkelä et al., 1996).

For measuring particle number size distribution in size range from 7 nm to $1\mu m$ we used Differential Mobility Particle Sizer (DMPS). The instrument consists of two parts: Differential Mobility Analyzer (DMA), made at the University of Helsinki, and Condensation Particle Counter (CPC), A10 manufactured by Airmodus Oy. The aerosol sample is neutralized using an X-ray source (Hamamatsu, Japan). DMPS was described in detail by Aalto et al.

125    (2001).

Particle size distributions ranging from 3 nm to 0.2 μm at the Fonovaya station are measured routinely using a Diffusional Particle Sizer (DPS). DPS consists of the Novosibirsk-type eight-stage screen diffusion battery (Reischl et al., 1991; Ankilov et al., 2002) connected to the Condensation Particle Counter (CPC). CPC Model 5.403 (GRIMM Aerosol Technik, Germany) was used until July 2019, and after – CPC Model 3756 (TSI Inc., USA).

Additionally, the distribution of particles within the size range of 0.3 μm to 20 μm (across 15 size bins) is measured using the Grimm aerosol spectrometer Model 1.108 (OPC).

Continuous measurements of different atmospheric parameters are concurrently performed. The measured parameters are meteorological, such as atmospheric pressure, temperature, relative humidity (RH), wind speed and direction, and global solar radiation. The trace gas concentrations were measured with a set of trace level monitors

indicated in Table 1 for $SO_2$, $O_3O_3$, and $NO_x$.

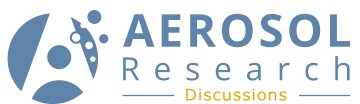

## 2.3. Data analysis

### 2.3.1. Classification of new particle formation events

We classifiedclassified all days based on NPF characteristics using two different methods: traditional manual classification, described by Dal Maso et al.. (2005)) and nanoparticle ranking analysis, described in Aliaga at al. (2023), and then compared the results obtained from these two approaches.

#### 2.3.1.1. Traditional new particle formation event classification

We classified NPF events, non-events and undefined events using the algorithm, described in Dal Maso et al. (2005). As this method was widely used in previous studies, here we call it "traditional". As mentioned before, we classify individual measurement days into three categories and calculate the fraction of days when NPF events occur, non-event days and undefined days. We analyze data visually on a day-to-day basis. Days, when new particle mode appears in sub-5 nm range and shows subsequent signs of growth longer than 2 hours, we classify as NPF event days. Days, when no new mode is observed, or if the new mode persists shorter than half of an hour, are classified as nonevent days. Other days are classified as undefined. If the month has less than 80% of data available, it is excluded from monthly statistics. We considered years from 2016 to 2021. We used DPS particle number size distribution for time period from January 2016 till June 2019. For the time period from July 2019 to November 2021 we used the distributions derived from NAIS (particle operation mode).

Traditional classification has several disadvantages. NPF events with weak intensity can be classified incorrectly due to instrumental limitations. In addition, when differently visualized, even non-event days clearly demonstrate signs of growth of the aerosol particles similar to NPF days (Kulmala et al., 2012). That is why in this study we compare traditional classification with the results of nanoparticle ranking analysis, which fits better at recognizing the quiet new particle formation (Kulmala, et al., 2022).

#### 2.3.1.2. Nanoparticle ranking analysis

We used nanoparticle ranking analysis to determine the occurrence probability and estimate the strength of NPF events. This method was described in Aliaga et al. (2023). Unlike the traditional classification, nanoparticle ranking analysis is objective, quantifiable, replicable and doesn't contain human bias. It is based on analysis of the particle number concentration at sizes from 2.5 nm to 5 nm. Particles of this range are sensitive to the presence of atmospheric NPF, and the increasing particle number concentration indicates nucleation and growth in the atmosphere. To perform ranking analysis, we extract the time series of the particle number concentration in the above-mentioned size range and filter the data (rolling median with 2 h window). The metric used is $\Delta N_{2.5-5}$ which srepresents the difference between the daily maximum and daily background concentrations of particles in this size range.. Then, each day is ranked according to isthis metric. For each measurement day, we have a single



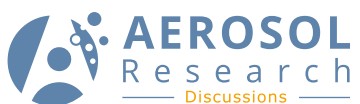

representative value of $\Delta N_{2.5-5}$ that allows us to compare results of nanoparticle ranking analysis with traditional NPF classification. All the days were grouped into 5% intervals based on their ranking to determine the corresponding potential NPF pattern for each interval. We used $\Delta N_{2.5-5}$ representing the peak daytime number
concentration of the formed particles with respect to the background concentration on that day, to see how it correlates with different parameters linked to NPF, such as trace gases concentrations, meteorological parameters, global solar radiation, condensation sink, etc. For calculations, we used data from NAIS from July 2019 to November 2021. For all atmospheric parameters we took daily medians between 10:00 and 14:00 local time because NPF events at Fonovaya station occur in this time interval. The raw $SO_2$ data has an increasing linear trend that is related
to instrument calibration. We corrected for this instrumental bias by subtractingsubtracting the trend line's slope from the measured concentrationsconcentrations during 2016 – 2021.

**2.3.2. Particle loss parameters**

Condensation sink (CS) and coagulation sink (CoagS) eCondensation sink is a parameter that shows how fast the molecules are lost by condensation onto pre-existing aerosol particles (Pirjola et al., 1999), and it is calculated from
the particle number size distribution. We calculated CS using two different methods. Firstly, it was calculated using particle number size distribution data from DPS and OPC. This dataset covers the period from January 2020 to the end of June 2021. The ranges of particle diameters covered by those instruments do not overlap, the data from 200 nm to 300 nm is missing, that is why the missing part was gapfilled with the nearest neighbor method (Ezhova et al., 2018). Secondly, CS was calculated using the data from DMPS. This dataset includes data from March 2020 to
September 2020 and from January 2021 to May 2021. The scatter plot comparing the results from the different instruments is shown in Fig. 1. The CS from both datasets are strongly correlated (Fig. 1). We relied on DMPS-based CS, calculated from the non-gapfilled distribution and corrected the DPS+OPC obtained values of CS. DPS+OPC data set has a longer data coverage, therefore the corrected CS values from this instrument were used in this study.

Coagulation sink is the parameter that shows how fast the particles of the certain size are lost by collisions with particles of larger sizes (Dal Maso et al., 2002). It is related to CS (Kulmala et al., 2012) and can be calculated using the following equation:

$$\text{CoagS}_{d_p} = \text{CS} \cdot \left(\frac{d_p}{0.71}\right)^m \tag{1}$$

where the exponent $m$ depends on the shape of the size distributions and approximated to be equal to -1.7 (Lehtinen
at al., 2007).

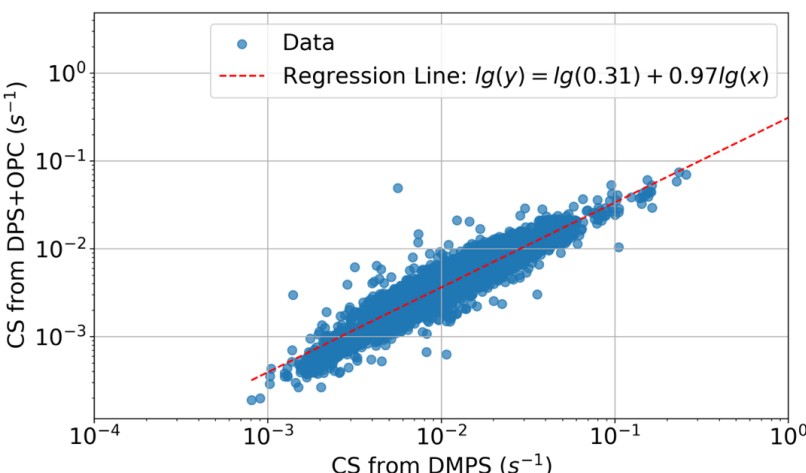

**Figure 1. Comparison between CS calculated from DMPS and DPS + OPC, hourly resolution. Correlation coefficient = 0.90**

### 2.3.3. Particle fformation and growth rates

Growth rate (GR) is a parameter that characterizes how fast the population of particles with diameter $d_p$ grows in time:

$$\text{GR} = \frac{dd_p}{dt} = \frac{\Delta d_p}{t} = \frac{d_{p2} - d_{p1}}{t_2 - t_1} \tag{2}$$

where $d_{p1}$ and $d_{p2}$ are the representative particle diameters at times $t_1$ and $t_2$ respectively (Kulmala et al., 2012).

In this study, growth rate (GR) values were calculated using the appearance time method, as described by Lehtipalo et al. (2014). This method involves selecting a time interval during which particles reach a specific size and calculating the GR based on the time difference between successive sizes. To do this, we select various particle 205 diameters and fit the time-dependent concentration of particles at each diameter with a sigmoid function. The time at which the sigmoid function reaches 50% of its maximum value is recorded for each diameter. Finally, the relationship between particle diameter and time is fitted with a linear function, and the slope of this line provides the GR value.

For calculating GR, we used the data from NAIS. GR were calculated using ion size distributions in the following 210 size ranges: from 2 to 3 nm, from 3 to 7 nm and from 7 to 20 nm. We used ion data for calculations because ion mobility range corresponds to a wider mobility diameter range than particle data. It is especially important when considering GR of particles with smaller diameters.



The formation rate ($J$) of particles in a size range between $d_p$ and $d_p + \Delta d_p$ was calculated as follows (Kulmala et al., 2012):


$$\frac{dN_{dp}}{dt} = \text{production} - \text{losses} = J_{dp} - \text{losses} \tag{3}$$

Particle formation rate at size $d_p$ is $J_{dp}$ in this equation. J can be estimated using the following equation:

$$J_{dp} = \frac{dN_{dp}}{dt} + \text{CoagS}_{dp} \cdot N_{dp} + \frac{GR}{\Delta d_p} \cdot N_{dp} + S_{\text{losses}} \tag{4}$$

We calculated J values using two different methods. In the first method, J values only for NPF event days were calculated using the NAIS data. Particle formation rate for 3-nm particles ($J_3$) was calculated using particle data,

meanwhile for $J_2$ we used ion data because in the ion mode, the detection limit is lower than in particle mode.

For calculating J, we take the time $t_1$, where the particles start forming, and time $t_2$, where new-formed particles grow till 6 nm. Thane we calculate the daily J time series and calculate the median J from $t_1$ to $t_2$. This value is a sought J used in this analysis.

In the second method, we used combined data from NAIS and DMPS, and calculated J values for all available days,

including NPF event days, non-event days and undefined. This method is fully automated. First method of J calculation gives better accuracy, and second method is needed for overall picture because it allows to calculate J values also for non-event days. For calculating GR and J, we used data from July 2019 till November 2021.

**2.3.4. Sulfuric acid proxy**

A simple sulfuric acid proxy was calculated from the parameterization introduced by Petäjä et al., 2009:

$$[\text{H}_2\text{SO}_4]_{\text{proxy}} = k \frac{[\text{SO}_2]\text{GlobR}}{\text{CS}} \tag{5}$$

where [$SO_2$] is the measured concentration of sulfur dioxide, GlobR is the measured global radiation and CS is the condensation sink. Parameter $k = 1.4 \cdot 10^{-9}$ m$^2$W$^{-1}$s$^{-1}$ was calculated for spring 2021 at the Fonovaya station based on the measurements with CI-APi-TOF (Garmash et al., 2024).

**3. Results and discussion**

**3.1. New particle formation event classification**

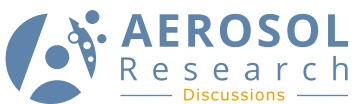

### 3.1.1. Traditional NPF event classification: formation and growth rates during NPF days

The classification of NPF events following Dal Maso et al. (2005) is illustrated in Fig. 2a. The fraction of NPF event days has maxima in spring (March 2020, April 2020) and autumn (October 2019, September 2020). This result is qualitatively similar to previous results for Fonovaya station (Lampilahti et al. 2023) and other boreal forest stations, such as SMEAR II (Dada et al., 2017). However, the year 2020 was unique in comparison to other years (Garmash et al., 2024). In 2020, 24% of days were classified as event days, 31.6% were undefined days, and 44.4% were nonevents, which differs strikingly from previous results. During 2016-2018, less than 10% of the days contained events, 21.1% were undefined, and 69% were nonevents (Lampilahti et al., 2023). The number of event days in 2020 is thus significantly higher than during 2016-2018 (Fig. 2b), especially in spring. The number of undefined days was also higher. Garmash et al. (2024) hypothesized that in spring 2020 warmer temperatures triggered early biogenic activity which caused a high NPF frequency in early spring (March-April). Not only spring, but also winter 2020 was exceptionally warm. Fig. 1b shows that more NPF events occurred also in October 2019 than in other years, which preceded the heatwave in 2020 (Garmash et al., 2024).

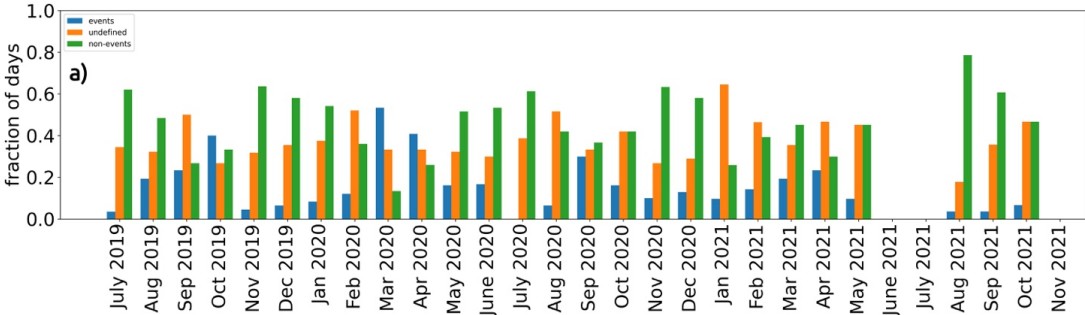

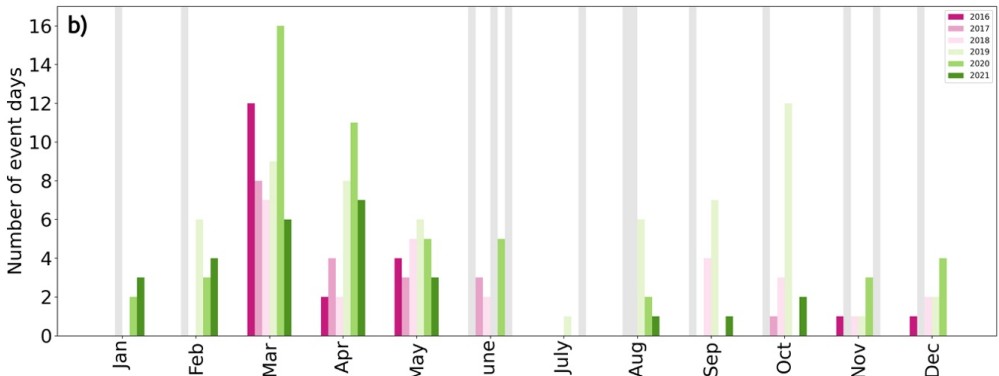

**Figure 2. a) Monthly traditional NPF event classification from July 2019 till October 2021, the y-axis representing fractions of NPF event, nonevent and undefined days. b) Number of NPF event days for each month shown for each year from 2016 to 2021. Gray shading corresponds to the months with data excluded from analysis (<80% data available).**


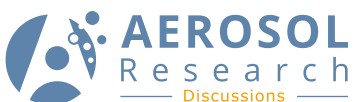

For the NPF event days, we calculated J and GR values and considered yearly (Table 2) and monthly medians for each

diameter range. Fig. 3a shows the boxplot of monthly $J_{2, ions}$ for the whole dataset. The median $J_{2, ions}$ for the whole measurement period equals to 0.01 cm$^{-3}$ s$^{-1}$. Fig. 3b represents the boxplot of monthly $J_{3, total}$. Yearly medians for $J_{2, ions}$ and $J_{3, particles}$ are listed in Table 2. J values have a seasonal variability according to fig. 3: the highest J values are observed in spring, followed by autumn. Summer and winter have the lowest median J. This result agrees with previous studies. J values for Fonovaya station were previously reported by Nieminen et al. (2018) and they were calculated for particles from 10 to 25

nm using the DPS data. The median J values were 1.2 cm$^{-3}$ s$^{-1}$ for spring, 0.7 cm$^{-3}$ s$^{-1}$ for summer, 1.0 cm$^{-3}$ s$^{-1}$ for autumn and 0.3 cm$^{-3}$ s$^{-1}$ for winter, so that the seasonal pattern is similar to our results. The same pattern was observed at another boreal forest site SMEAR II station in Hyytiälä, Finland. J values reported by Nieminen et al., (2018) from SMEAR II have similar seasonal variabitily. J values from 5 to 30 nm for Fonovaya station were calculated by Lampilahti et al. (2023). The median value was equal to 0.8 cm$^{-3}$ s$^{-1}$.

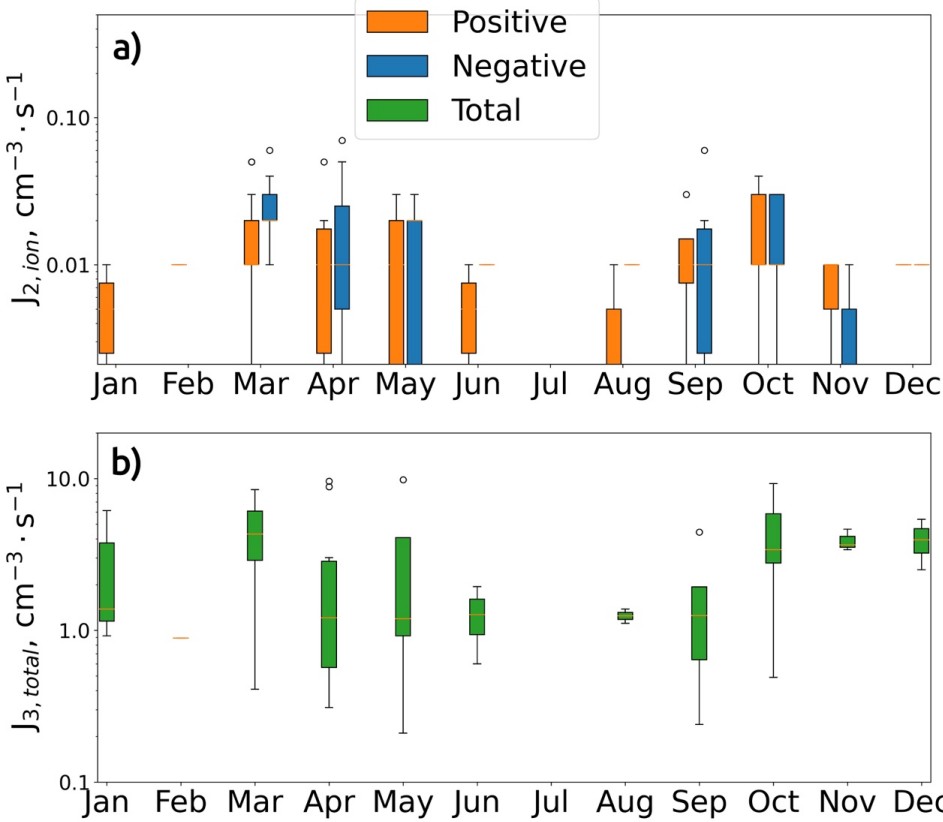


**Figure 3. Monthly boxplots for formation rates. y-axis represents particle formation rates (J), x-axis represents months. Positively charged ions (3a) are marked with orange and negatively charged marked with blue. Total values (positively charged + negatively charged) are marked as green.**




Figs. 4a, 4b, and 4c show the monthly median total GR values (calculated from positive + negative ions) in the size ranges

2-3, 3-7 and 7-25 nm, respectively. For the size ranges 3-7 and 7-25 nm (Fig. 4b and c), we can clearly see a seasonal variability: monthly median values have minima in winter and maxima in May. For summer there is not enough measurement data for drawing any firm conclusions.

For Fonovaya station, GR were reported by Lampilahti et al. (2023). In that study, growth rates were calculated in the

diameter range from 5 to 20 nm, and the median value of GR was equal to 2.0 nm h$^{-1}$ during 2016-2018. This value is lower than the values we got in the current study (Table 2, the closest variable is GR$_{7-20}$, that is equal to 2.9 - 3.3 nm h$^{-1}$ depending on the year). The difference can be caused by several reasons: first, we use ion size distribution for calculations, whereas in previous study the particle size distribution was used; second, we used NAIS data instead of DPS data; and third, we used different methods for GR calculations (appearance time versus mode fitting method in Lampilahti et al. (2023)). The observed

growth rates reported for various boreal forest sites in the literature vary from about 0.5 nm h$^{-1}$ to 5.3 nm h$^{-1}$ (5[th] to 95[th] percentile values), with a median GR of 2.7 nm h$^{-1}$ (Kerminen et al., 2018) At the SMEAR II station in Hyytiälä, Finland, the median values of GR were found to be the highest in summer (4.5 nm h$^{-1}$) and the lowest in winter (2.0 nm h$^{-1}$) (Nieminen et al., 2018). For the same SMEAR II station, Yli-Juuti et al. (2011) reported the following median GR values: 1.9 nm h$^{-1}$ for the size range from 1.5 to 3 nm, 3.8 nm h$^{-1}$ for the size range from 3 to 7 nm, and 4.3 nm h$^{-1}$ for the size

range from 7 to 20 nm. That research covered the time period 2003-2009. Overall both the seasonal pattern of GR (Fig. 4) and its size dependency (Table 2) observed in our study are broadly in line with earlier studies in various boreal forest environments.

**Table 2. Yearly medians of formation and growth rates. J$_2$ is calculated using NAIS ion data, and J$_3$ was calculated using NAIS particle data.**

| | J$_2$ pos, | J$_2$ neg, | J$_3$ pos, | J$_3$ neg, | GR$_{2-3}$ pos, | GR$_{2-3}$ neg, | GR$_{3-7}$ pos, | GR$_{3-7}$ neg, | GR$_{7-20}$ pos, | GR$_{7-20}$ neg, | (J$_{2\,pos}$ + J$_{2\,neg}$)/J$_{3\,total}$ |
|---|---|---|---|---|---|---|---|---|---|---|---|
| | $cm^{-3} \cdot s^{-1}$ | $cm^{-3} \cdot s^{-1}$ | $cm^{-3} \cdot s^{-1}$ | $cm^{-3} \cdot s^{-1}$ | $nm \cdot h^{-1}$ | $nm \cdot h^{-1}$ | $nm \cdot h^{-1}$ | $nm \cdot h^{-1}$ | $nm \cdot h^{-1}$ | $nm \cdot h^{-1}$ | |
| 2019 | 0.01 | 0.01 | 1.8 | 2.1 | 1.5 | 2.3 | 2.3 | 2.2 | 4.2 | 3.3 | 0.01 |
| 2020 | 0.01 | 0.01 | 1.1 | 1.3 | 1.6 | 1.1 | 2.7 | 3.3 | 3.3 | 3.3 | 0.02 |





| 2021 | 0.01 | 0.01 | 0.8 | 0.6 | 0.8 | 1.2 | 2.0 | 1.5 | 3.3 | 2.9 | 0.03 |
|---|---|---|---|---|---|---|---|---|---|---|---|


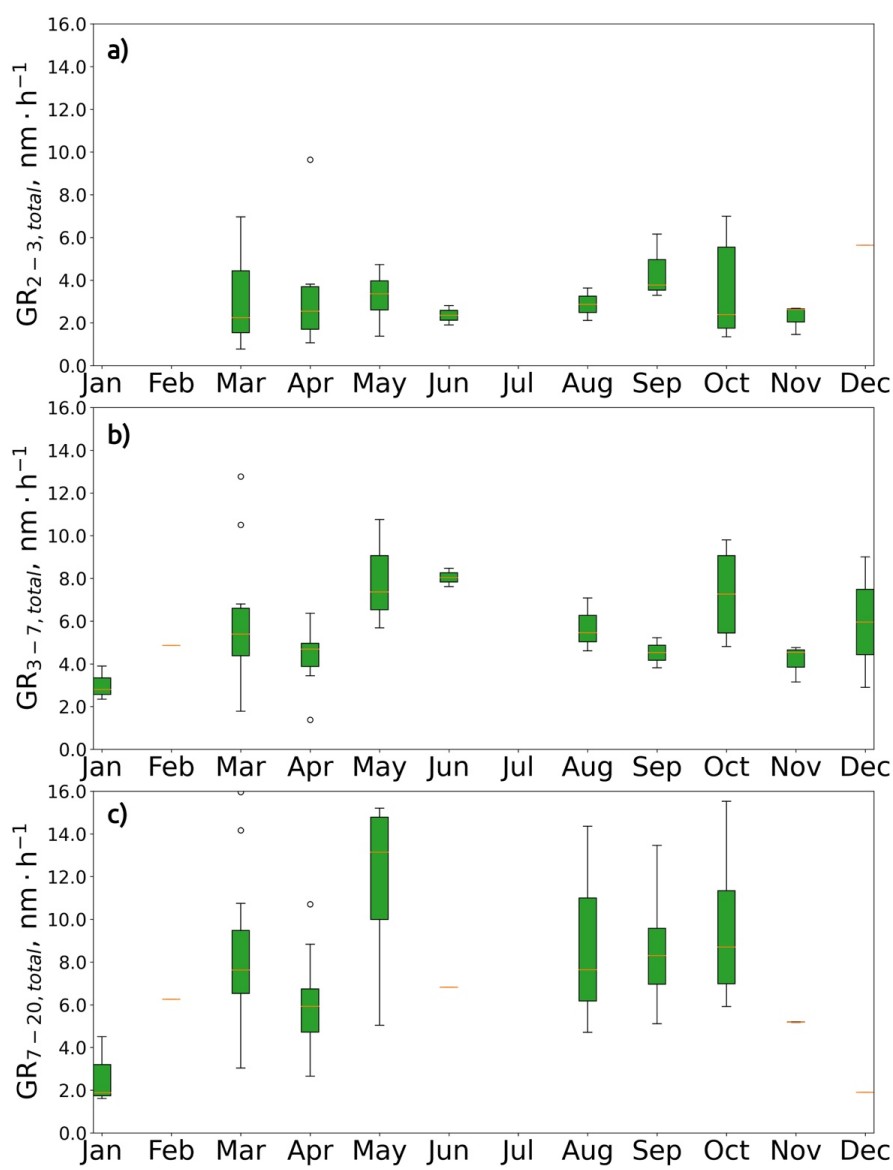

Figure 4. Monthly boxplots for growth rates. Y-axis represents the total GR values, x-axis represents months.




Previously, GR for Fonovaya station was reported by Nieminen et al., 2018. In that study, the authors calculated GR from 10 to 25 nm for 36 different measurement sites all over the world. The median value of GR at the Fonovaya station was the

highest in summer ($6.7 \, \text{nm} \, \text{h}^{-1}$) and the lowest in winter ($0.8 \, \text{nm} \, \text{h}{-1}$), while the corresponding medians across all the stations were equal to  $4.0 \, \text{nm} \, \text{h}^{-1}$  and $2.9 \, \text{nm} \, \text{h}^{-1}$. In comparison to other sites, the seasonal variability for Fonovaya station was higher. The spring median for Fonovaya was reported as $2.6 \, \text{nm} \, \text{h}^{-1}$, and autumn median as $2.3 \, \text{nm} \, \text{h}^{-1}$ (Nieminen et al., 2018). For calculations, the authors used DPS data. In our study, we also observe a similar seasonal variability: $GR_{7-20}$ values are lower in winter and increased in May (Fig. 4c).

**3.1.2. Nanoparticle ranking analysis and comparison to traditional classification**

In order to have a quantifiable parameter that characterizes NPF, we decided to perform nanoparticle ranking analysis. The first step of nanoparticle ranking analysis is extracting hourly particle concentrations in the 2.5 to 5 nm size range. We grouped the time series by season (Fig. 5) in order to understand how those values vary seasonally. Most of the NPF events

at the Fonovaya station fall on March and early April (Fig. 1a), and accordingly, in ranking analysis we observe the maximum concentration in spring at around 12:00 LT. Similar result is observed also at SMEAR II station, where spring maximum concentration is also reached at around 12:00 LT (Aliaga et al., 2023). Winter and autumn have very similar daily medians and profiles, while summer time concentrations are lower.

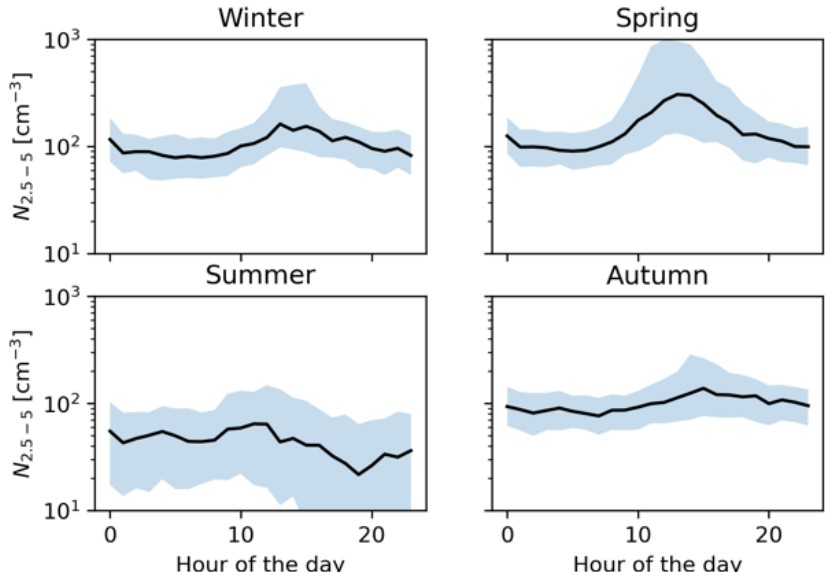

**Figure 5. Daily medians of particle concentration in 2.5 to 5 nm range grouped by season. x-axis represents the hour**

**of the day, y-axis is the particle concentration.**




Fig.6 shows daily median particle size distributions grouped into 5% intervals based on $\Delta N_{2.5-5}$ values. The figure illustrates the shape of particle distribution in each interval. In this figure, one can clearly see that smaller rank values visibly correspond to non-events (0 – 60% interval), whereas NPF events become visible for higher rank values (60% – 100% intervals).

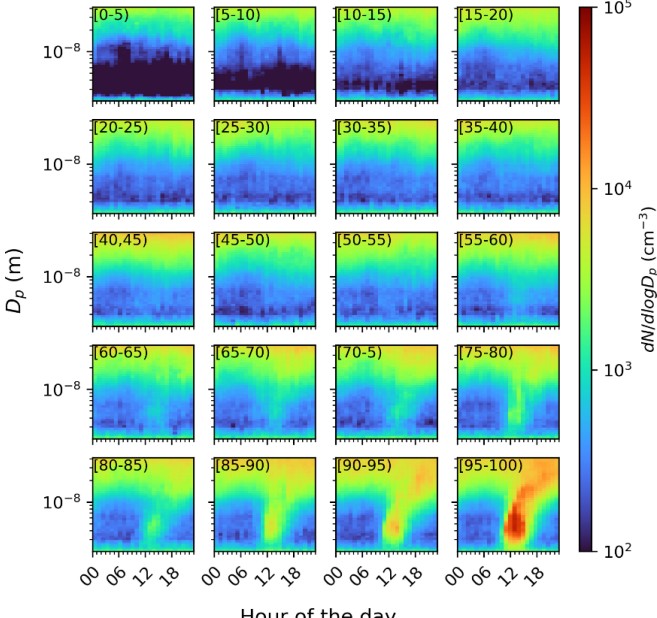

**Figure 6. Daily median number particle size distribution grouped into 5% intervals based on the $\Delta N_{2.5-5}$, as an illustration of the potential NPF events in each interval**

Furthermore, we compared results from the nanoparticle ranking analysis with traditional NPF event classification (Fig. 7). The histogram displays the percentile rankings divided into 5% bins, and the color-code represents the traditional classification. Days with ranks below 60% are mostly classified as non-event days or undefined events. Above 60% interval, the number of non-event days decrease, and at the highest interval, 95-100%, non-events are not observed. The fraction of days classified as NPF event days starts to grow after percentile ranking of 85% and above and reaches the maximum at 90% - 100% intervals. At the interval 60-85%, weak NPF events are visible. This result goes in line with the results presented by Aliaga at al. (2023), where the similar relationship between the results of ranking analysis and traditional NPF classification was observed for the SMEAR II station. Ranks below 65% are classified as non-event days, from 65% to 85% NPF events are weak, above 85% NPF events are clear with maximum intensity at 90-100% interval. It helps to identify $\Delta N_{2.5-5}$ corresponding to traditionally classified NPF and non-NPF events and will be used in the next section. Note, however, that the present analysis was performed on the data set containing the exceptional year 2020 with a large number of NPF events, which may have influenced the comparison between $\Delta N_{2.5-5}$ and NPF events at higher $\Delta N_{2.5-5}$ values.

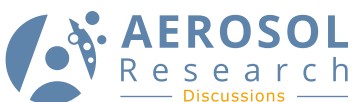

Fig. 8 represents the correlation between $J_3$ and $\Delta N_{2.5-5}$, and those parameters have a strong positive dependence. We did a statistical test, and the correlation is statistically significant for all the seasons (Table 3). Similar results for the SMEAR II station were published by Aliaga et al. (2023), where daily maximum $J_3$ also correlated clearly with the $\Delta N_{2.5-5}$.

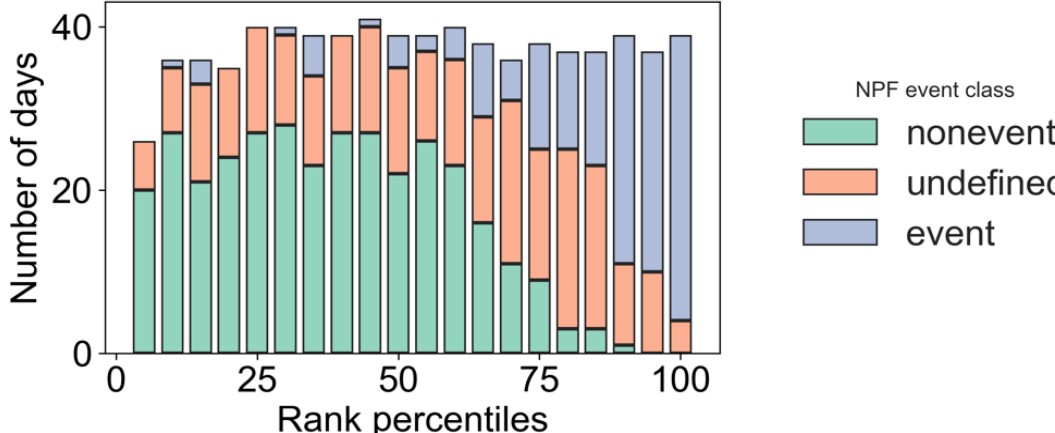

**Figure 7. Comparison between percentile ranking and traditional classification, with nanoparticle rank percentiles on x-axis and number of all days within a given rank on y-axis, and traditional NPF classes marked with color**.

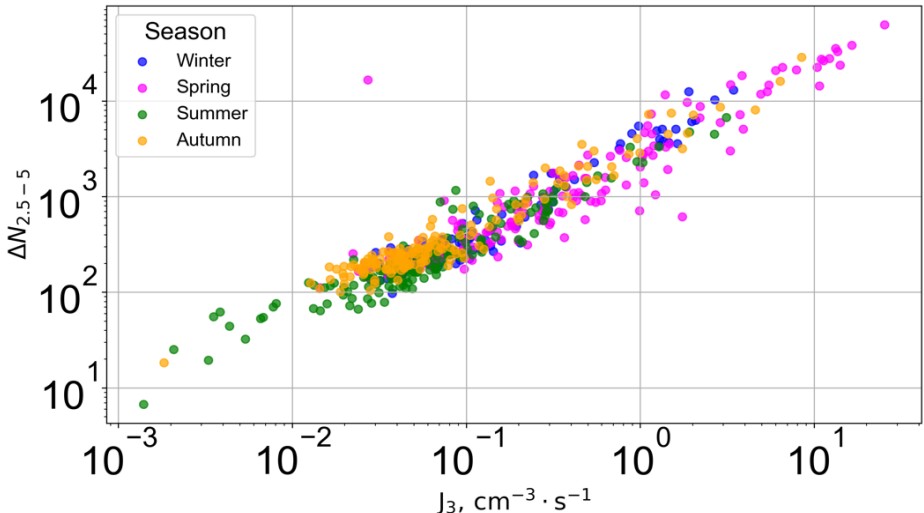

**Figure 8. Correlation between $J_3$ values on x-axis and $\Delta N_{2.5-5}$ in 2.5 to 5 nm range on y-axis. Different seasons are marked with colors.**

**3.1.3. Correlations between nanoparticle ranking and different atmospheric parameters.**



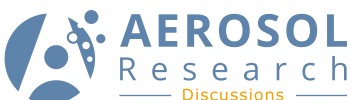

Using nanoparticle ranking framework, we can investigate the influence of different atmospheric parameters on NPF occurrence. We considered the correlation between $\Delta N_{2.5-5}$ and relevant atmospheric variables, such as concentrations of trace

gases ($SO_2$, $O_3$, $NO$, $NO_2$), global solar radiation, temperature, relative humidity (RH) and wind speed. $\Delta N_{2.5-5}$ values that correspond to percentiles above 85% are associated with NPF events and those below 40% with nonevents (Fig. 7), and the corresponding $\Delta N_{2.5-5}$ values are above 2400 $cm^{-3}$ for NPF events and $\Delta N_{2.5-5}$ below 250 $cm^{-3}$ for non-events. The correlations are shown in Fig. 9 with all the data points color-graded seasonally. The light blue shadow in those plots indicates the values of $\Delta N_{2.5-5}$ corresponding to a high probability (percentile >85%) of NPF event days, and the green shadow highlights the days

with low NPF probability (percentile <40%).

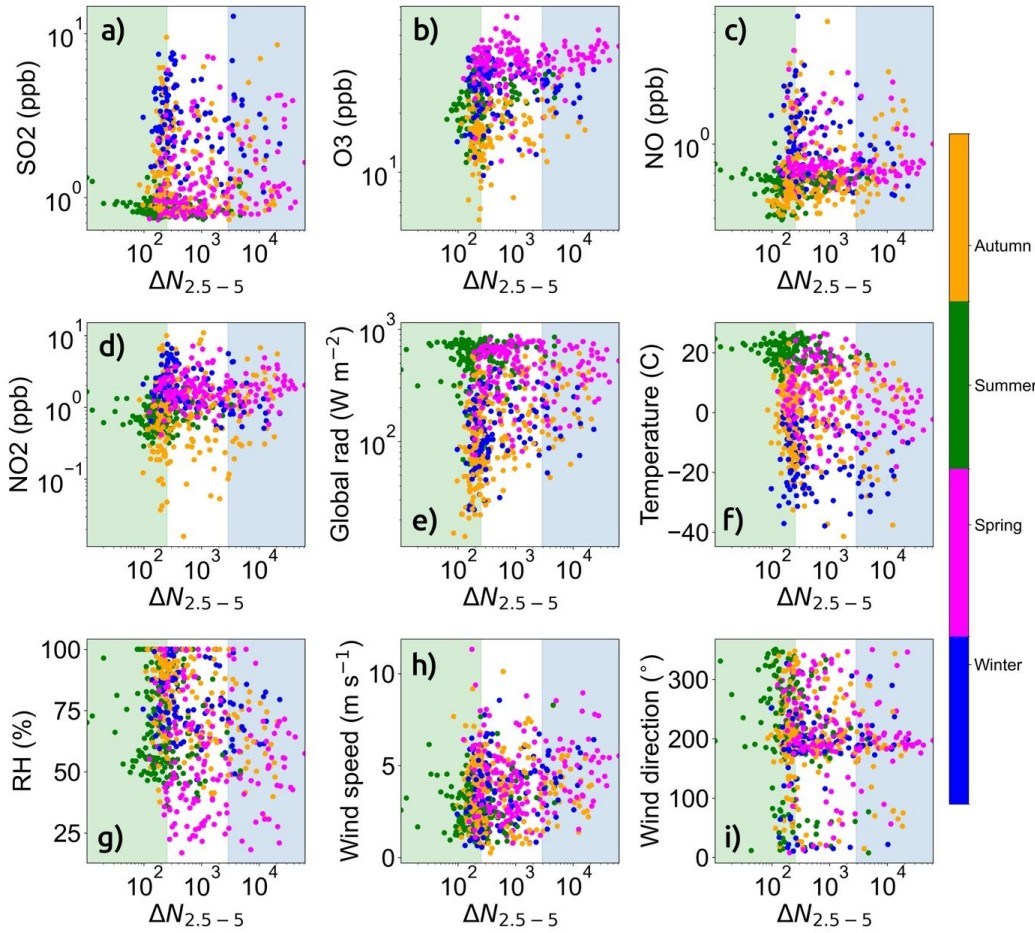

**Figure 9. Correlations between $\Delta N_{2.5-5}$ and atmospheric variables: a) $SO_2$ concentration, b) $O_3$ concentration, c) NO concentration, d) $NO_2$ concentration, e) Global solar radiation, f) Temperature, g) Relative humidity, h) Wind speed, and i) Wind direction on y-axis. Blue shadow highlights the area with the maximum number of event days (above 85% percentile), green shadow is showing the area with the maximum number of nonevent days (below 40%**

**percentile). Colors of the symbols represent different seasons.**





|  | R value winter | R value spring | R value summer | R value autumn |
|---|---|---|---|---|
| SO₂ | 0.041 | 0.208 | -0.136 | 0.044 |
| O3 | 0.145 | 0.23 | 0.346 | 0.194 |
| NO | -0.013 | 0.044 | 0.034 | 0.18 |
| NO₂ | -0.166 | 0.139 | 0.222 | 0.129 |
| GlobRad | 0.382 | 0.158 | 0.042 | 0.422 |
| Temperature | 0.396 | -0.213 | -0.226 | 0.225 |
| RH | -0.472 | -0.194 | -0.138 | -0.567 |
| Wind speed | 0.169 | 0.149 | 0.126 | 0.04 |
| H₂SO₄ proxy | 0.493 | 0.224 | 0.124 | 0.286 |
| CS | -0.455 | -0.009 | -0.083 | 0.109 |
| J₃ | 0.964 | 0.917 | 0.946 | 0.964 |

**Table** 3**. Pearson's correlation coefficients (R) between $\Delta N_{2.5-5}$ and different atmospheric parameters for each season. We marked in the table the R-coefficients for which the correlations are statistically significant (p<0.05). For SO₂, O₃, NO, NO₂, global solar radiation, H₂SO₄ proxy, CS and J₃ we did a significance test using log10(parameter) and log10($\Delta N_{2.5-5}$). We did not apply log transformation to temperature, RH and wind speed.**

Under the influence of solar radiation, SO₂ in the atmosphere is oxidized by OH to form sulfuric acid (H₂SO₄) vapor which plays a central role in aerosol formation and growth because its low volatility and high affinity for water makes it a key component in cluster formation and early growth of such clusters (Petäjä et al., 2009, Kulmala et al., 2013, Cai et al., 2021). The correlation between $\Delta N_{2.5-5}$ and SO₂ is positive and statistically significant in spring (Table 3), where the most NPF events are observed (Fig. 2b). Similarly, Lampilahti et al. (2023) reported that median values of SO₂ concentration have a

statistically significant difference between event days and non-event days at the Fonovaya station during spring 2016-2018.

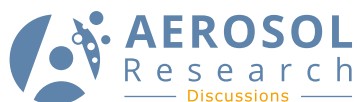

This study suggested that $SO_2$ is associated with anthropogenic emissions coming from the city of Novosibirsk and Kazakhstan, and the median $SO_2$ concentration at the Fonovaya is about an order of magnitude higher than at the SMEAR II station. High $SO_2$ concentrations at SMEAR II station are also associated with anthropogenic emission sources in St. Petersburg, Baltic countries, and Kola Peninsula (Hulkkonen et al., 2012, Riuttanen et al., 2013). The highest $SO_2$

concentrations were observed in winter (Fig. 9a), because in Siberia house heating is done by burning fossil fuels (coal, oil, gas) that releases $SO_2$ into the atmosphere, but also because of airmass transport from polluted areas. During this season, however, the amount of solar radiation is low, which is possibly why sulfuric acid concentration is low (Fig. 10b) and the number of NPF events is also low. Similar seasonal patterns were reported for the SMEAR II station: $SO_2$ concentration has a maximum in winter (February), and the lowest levels prevail from May till September. The winter maximum is connected

to heating and slower atmospheric chemistry due to low intensity of sunlight (Nieminen et al., 2014). The connection between $SO_2$ concentration and NPF frequencies in previous studies is ambiguous, as the NPF frequencies were reported to have either higher (Birmili and Wiedensohler, 2000, Woo et al., 2001, Dunn et al., 2004, Boy et al., 2008, Young et al., 2013, Zhao et al., 2015) or lower (Wu et al., 2007, Dai et al., 2017) concentration depending on the location. One study reported that the correlation between NPF occurrence and $SO_2$ concentration depends on the season: in spring and summer $SO_2$ concentrations

during the NPF event days were higher than during the non-event days (Qi et al., 2015).

Fig 9b shows the dependence between $\Delta N_{2.5-5}$ and $O_3$ concentration. The correlation is statistically significant in spring and summer (Table3), and ozone has a seasonal pattern with maximum in spring and minimum in autumn. A similar seasonal pattern for ozone was observed at the SMEAR II station, the concentrations being the highest in spring (March – April) and lowest in early winter (November) (Chen et al., 2018), as well as at SMEAR Estonia (Noe at al., 2015). Such behaviour is

connected to the spring recovery of photochemical production (Dibb et al., 2003) and ozone accumulation during winter (Liu et al., 1987). The ambient ozone concentration at the Fonovaya station was reported to be lower than at the SMEAR II and SMEAR Estonia stations (Lampilahti et al., 2023). That study also reported that the difference between ozone concentrations during NPF events and nonevents is statistically significant, with higher ozone concentrations during NPF event days. The relation between the ozone concentration and NPF occurrence has been studied before, and $O_3$ is expected to enhance NPF

because it is an oxidant forming extremely low volatility organic compounds (ELVOC) (Donahue et al., 2012, Ehn et al., 2014). Other studies also considered ozone to have positive influences on NPF (Woo et al., 2001, Berndt et al., 2006). In contrast, Carnerero et al. (2019) showed that at a site in Spain, higher ozone concentrations were associated with lower NPF occurrences, but this correlation may not be causal due to associations with other atmospheric parameters, such as temperature, RH, or global solar radiation. Another reason for positive $O_3$ correlation with NPF could be due to the enhanced

ozone production during VOC oxidation in the presence of $NO_x$ which is associated with pollution and, hence, higher $SO_2$ and sulfuric acid as well (Bousiotis et al., 2021). At this Siberian site, NPF occurs predominantly within polluted air masses (Lampilahti et al., 2023, Garmash et al., 2024).

NO and $NO_2$ concentrations remain relatively constant during the spring season for all values of $\Delta N_{2.5-5}$ (Fig. 9c, d). The positive relationship between NO and $\Delta N_{2.5-5}$ is statistically significant in autumn (Table 3). With $NO_2$, the relationship is

statistically significant in winter (negative correlation) and summer (positive correlation). High $NO_x$ concentrations are associated with pollution: for instance, at the SMEAR II station higher NOx was associated with air mass transport from polluted areas (Riuttanen et al., 2013). In addition, NO can also be emitted from the soils (Kesik et al., 2005, Pilegaard et



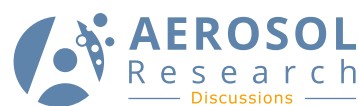

al., 2013). From fig. 9c,d we can see that $NO_x$ concentrations are the highest during winter, spring, and autumn. This follows the observations at the SMEAR II station, where $NO_x$ concentrations are highest during winter months and early spring

because of combustion sources and weakness of photochemical sink (Riuttanen et al., 2013). NO reacts with ozone and volatile organic compounds (VOCs) emitted by vegetation (Wildt et al., 2014). $NO_x$ can affect NPF occurrence in different ways: it can reduce NPF because VOC oxidation in presence $NO_x$ produces higher volatility products, but also $NO_x$ contributes to oxidant recycling (Sillman, 1999), and as a result this process can suppress NPF. The influence of $NO_x$ on NPF was studied in the laboratory chamber by Yan et al. (2020), revealing that $NO_x$ suppresses NPF, but the suppression effect is

nonuniform- and particle size-dependent. A similar dependence of NPF on $NO_x$ was reported by Zhao et al. (2018). Other findings (Wildt et al., 2014) indicate that $NO_x$ can either promote or inhibit NPF depending on its concentration levels and the availability of other atmospheric components like VOCs and $SO_2$. Specifically in their experiments, at $NO_x$ concentrations above 2 ppb, the particle formation rate decreased by up to 75% compared to $NO_x$-free conditions. For the Fonovaya station in spring, previous results showed that $NO_x$ concentrations are higher than at other boreal forest sites, and that the difference

in concentrations between the NPF events and non-events is statistically significant with higher $NO_x$ concentrations during NPF events (Lampilahti et al., 2023). NPF in Siberia is most likely driven by anthropogenic pollution, so $NO_2$ emissions can influence NPF occurrence.

Global solar radiation (Fig. 9e) is one of the most important factors for the occurrence of NPF (Kerminen et al., 2018), primarily because it initiates the chemical reactions that contribute to aerosol formation in the atmosphere. Aaltonen et al.

(2011) highlighted that high levels of solar radiation can enhance the photochemical reactions that lead to the production of oxygenated organic compoundsas well as oxidize $SO_2$, increasing $H_2SO_4$ concentrations in the atmosphere (Petäjä et al., 2009), which is essential for nucleation and growth of new particles. In our study, the correlation between $\Delta N_{2.5-5}$ and global solar radiation is positive and statistically significant in winter, spring, and autumn (Table 3). Previously we found out that at the Fonovaya station, the biggest fraction of NPF events take place during clear-sky or low-cloudiness conditions

(Lampilahti et al, 2023). A similar dependence was observed at the SMEAR II station (Dada et al., 2017). Our present analysis aligns well with previous studies, showing that higher values of $\Delta N_{2.5-5}$, associated with increased NPF occurrence, correspond to increased global radiation (Kanawade et al., 2014, Pierce et al, 2014, Qi et al., 2015, Wonaschütz et al., 2015).

The correlation between the temperature and $\Delta N_{2.5-5}$ (Fig. 9f) is negative and statistically significant in spring and summer, and negative but not statistically significant in autumn. The effect of temperature on NPF is ambiguous, and different studies

are showing different dependencies. Dada et al. (2017) found out that at the SMEAR II station NPF is more frequent during increased temperatures in cold season, and decreased temperatures during warm seasons. At Fonovaya, warmer seasons are spring and summer (Fig. 9f), and the correlations with $\Delta N_{2.5-5}$ are negative, which agrees with the results for SMEAR II. Also, Dada et al. (2017) found out that both very low (below -21°C) and very high (above 25°C) temperatures correspond to nonevent days. Bousiotis et al. (2021), explored the correlation between the temperature and NPF occurrence for various

sites worldwide. At most of the sites, temperature relationship with NPF was positive, but at several sites the correlation was negative. Different studies are showing different effect of temperature on NPF likely because temperature has both direct and indirect effects which can either enhance or suppress NPF (Kerminen et al., 2018). Increased temperatures in spring enhance biogenic emissions of aerosol precursor vapors and their oxidation to low-volatility vapors (Grote and Niinemets, 2008).



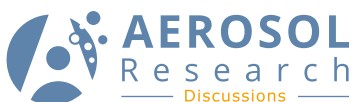

However, as shown in Garmash et al. (2024), early spring with low temperature is favorable to NPF compared to late spring,
which might be due to enhanced stability of molecular clusters at lower temperatures.

The relationship between RH and $\Delta N_{2.5-5}$ (Fig. 9g) is statistically significant in winter, spring and autumn, and the dependence is negative. Previous studies showed that that RH tend to be lower during NPF event days in comparison to non-event days (Birmili and Wiedensohler, 2000, Kanawade et al., 2014, Pierce et al., 2014, Qi et al., 2015, Zhao et al., 2015, Salma et al., 2016). The negative effect on NPF can be explained with with negative influence of RH on solar intensity and photochemical
reactions and  precursor vapors as a result (Hamed et al., 2011). A similar dependence was observed also at the SMEAR II station (Dada et al., 2017). Overall, our result agrees with previous studies.

Wind speed (Fig. 9h) has a positive and statistically significant correlation with $\Delta N_{2.5-5}$ in winter and spring (Table 3). According to Bousiotis et al. (2021), wind speed can have both positive and negative effect on NPF occurrence. A higher wind speed can promote NPF by increasing mixing and reducing CS, while on the other hand it can suppress NPF due to
increased dilution of condensing vapors. In general, the influence of wind speed on NPF was reported to be different for different sites (Bousiotis et al., 2021).

In addition, we considered the link between $\Delta N_{2.5-5}$, condensation sink (CS) and sulfuric acid proxy (Fig. 10a, b). CS is a very important parameter in atmospheric observations because it describes how fast precursor vapors are lost to aerosol surface and hence it is known as a factor that suppresses NPF (Kulmala and Kerminen, 2008). At the SMEAR II station, NPF occurs
during low values of condensation sink (Dada et al., 2017), and the CS has a seasonal pattern with a maximum in summer and a peak value in July, and with a minimum in around November (Nieminen et al., 2014). At the Fonovaya station, the seasonal CS pattern is different, with maximum values in winter and spring. In other studies, low CS sink is associated with increased NPF occurrence (Boy and Kulmala, 2002, Hyvönen et al., 2005, Baranizadeh et al., 2014).

**Figure 10. a) Correlations between $\Delta N_{2.5-5}$ on x-axis and CS, calculated using DPS + OPC data and corrected on the**

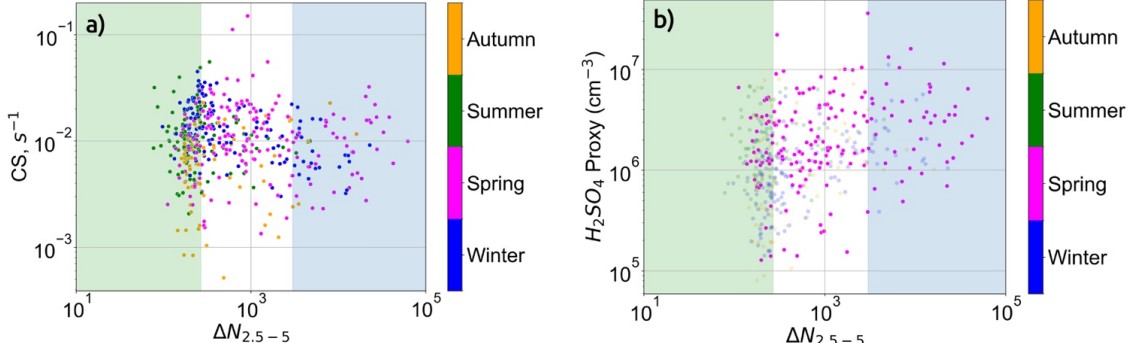

**cross-correlation coefficient on y-axis. Colors represent different seasons. b) Sulfuric acid proxy concentration on y-axis compared to the $\Delta N2.5-5$ on x-axis, colors represent different seasons. The proxy calculation is designed for spring, that is why all other seasons except spring are plotted as transparent.**





We compared the calculated sulfuric acid proxy to $\Delta N_{2.5-5}$ (Fig. 10b). The correlation is positive and statistically significant in winter, spring and autumn. $H_2SO_4$ is a precursor vapour for NPF, and a connection between those parameters was reported

in various studies (Petäjä et al., 2009, Paasonen et al., 2010, Wang et al., 2011, Yao et al., 2018). At the SMEAR II station, the $H_2SO_4$ proxy reaches a maximum in spring (March and April), and a minimum in autumn (Nieminen et al, 2014). A similar seasonal pattern is observed at the Fonovaya station (Fig. 10b). The seasonal variations in $H_2SO_4$ proxy is affected by the seasonal variations in the $SO_2$ concentration, CS, and global solar radiation. Nieminen et al. (2014) reported that $H_2SO_4$ concentration alone did not separate NPF event and non-event days, suggesting that oxidized organics also play an important

role in determining the occurrence of NPF. Other studies reported higher $H_2SO_4$ concentrations during NPF event days (Birmili et al., 2003, Boy et al., 2008). Our result agrees with those studies.

## 4. Conclusions

In this study, we investigated the NPF process and factors affecting it at the Fonovaya station in Siberia. We did a traditional NPF event classification using a 2-year-long dataset of NAIS measurements and compared NPF frequencies during six years

from 2016 to 2021. The results that we got follow in general previous studies: we observed the maximum number of NPF events in March and the second smaller peak in autumn; however, with abnormally high number of events during spring 2020 and autumn 2019. We also reported aerosol formation and growth rates calculated from the NAIS data. The growth rates are somewhat lower than at the SMEAR II station, but the numbers are comparable. We observed seasonal variability of particle formation rates J with a maximum in spring and autumn and a minimum in winter. Growth rates also have a seasonal

variability, with a minimum in winter and a maximum in May. The seasonal variability of GR at the Fonovaya station is larger than at the other boreal forest sites reported in the literature. By far this is the longest formation and growth rates dataset reported for the Siberian region.

We compared the results of traditional event classification with nanoparticle ranking method, which was used for Siberian data for the first time. NPF events occur mostly at percentile ranking above 85%. Percentile rankings below 40% correspond

mostly to non-events. We then investigated the dependence between $J_3$ and $\Delta N_{2.5-5}$, and the correlation was strongly positive and statistically significant for every season. This dependence illustrates the clear connection of $\Delta N_{2.5-5}$ with the probability and intensity of NPF.

Using nanoparticle ranking method, we studied how various atmospheric parameters influence NPF at the Fonovaya station. $SO_2$ plays an important role in NPF, and its influence is statistically significant in spring where most of the NPF events are

observed. $SO_2$ is oxidized with OH and form sulfuric acid vapor that plays a key role in aerosol formation and growth. It has seasonal variability with a maximum in winter possibly because of residential heating. However, in winter due to lack of solar radiation, less sulfuric acid is formed, and that is possibly why the $SO_2$ influence on NPF is statistically not significant. The correlation of ozone with $\Delta N_{2.5-5}$ is positive and statistically significant in spring and summer, and it has a seasonal pattern with a maximum in spring and a minimum in autumn. The influence of ozone on NPF can be explained by VOC

oxidation which enhances the occurrence of NPF. $NO_x$ plays a role in the NPF process at the Fonovaya station because particle formation in Siberia occurs mostly in polluted air masses. The relationship between NO and $\Delta N_{2.5-5}$ is statistically significant in autumn, and $NO_2$ has negative significant correlation to $\Delta N_{2.5-5}$ in winter and positive significant in summer.

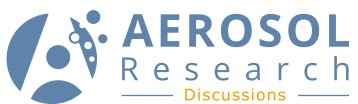

The influence of $NO_x$ on Siberian NPF is inconsistent. Global solar radiation is one of the most important factors for the occurrence of NPF and its influence is statistically significant in winter, spring and autumn - seasons when radiation can be

low. Solar radiation enhances NPF by starting photochemical reactions that increase the oxidation of VOC and $SO_2$ leading to increased concentrations of $H_2SO_4$ and low-volatility organic vapours  in the atmosphere. The effect of temperature on NPF is negative and statistically significant is spring and summer, so NPF is more frequent with decreased temperatures during warmer seasons, which agrees with results from SMEAR II. RH has a negative influence on NPF because of its connection to reduced solar intensity. RH connection to the concentration of small particles is strongest of all other variables

and statistically significant in winter, spring and autumn. CS suppresses NPF in Siberia, but its influence is statistically significant only in winter when CS reaches maximum values. One of the most important parameters for Siberian NPF is the $H_2SO_4$ concentration, and the correlation of NPF with sulfuric acid proxy is significant in winter, spring and autumn. Sulfuric acid promotes aerosol formation because of its low volatility and high affinity for water, influence both cluster formation and early growth of these clusters onto the growing particles. For further perspectives, future studies could focus on exploring

additional precursors and atmospheric parameters influencing NPF, and conducting comparative analysis with other boreal forest sites to better understand regional and global implications of Siberian NPF.

### Acknolegements

We acknowledge the following Projects: ACCC (The Atmosphere and Climate Competence Center) Flagship funded by the Academy of Finland Grant No. 357902, Academy professorship funded by the Academy of Finland (Grant No. 302958),

Business Finland project CARBON+, Academy of Finland mobility Grant Nos. 333581, 334625, INAR Project funded by Jane and Aatos Erkko Foundation, European Research Council (ERC) Project ATM-GTP (Atmospheric Gas-to- Particle conversion) Contract No. 742206, Novo Nordisk Foundation Start Package Grant (Grant number NNF24OC0090482).

### Author contributions

AL, OG, MA, DD, BB, TP, MK and EE organised the measurement campaign, AL, DA, JL contributed to data analysis, OG,

DA, MA, JL, VMK, TP, MK and EE contributed to scientific discussion, AL wrote the manuscript with the help of co-authors.

### Competing interests

At least one of the (co-)authors is a member of the editorial board of Aerosol Research.

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
