# Peer review of "Insights into new particle formation in Siberian boreal forest from nanoparticle ranking analysis"

_Aerosol Research, 2025_

## Author Response (AR1)

Dear Reviewer #1,

We sincerely thank you for your time and thoughtful comments on our manuscript titled *"**Insights into new particle formation in Siberian boreal forest from nanoparticle ranking analysis**"* submitted to *Aerosol Research*. We appreciate the reviewers' efforts to improve the quality of our work. We have carefully considered all the comments and revised the manuscript accordingly. Below, we provide a point-by-point response to each comment. All the changes in the manuscript are highlighted in green color.

1. There are occasional grammatical or typographical issues (e.g., duplicated words like "confirmingconfirming", "wewere", "thethe"), which should be corrected for improved readability.

We apologize for the inconvenience for reading caused by these typos. We have read through the text thoroughly and corrected the typos and grammatical issues we were able to find in the manuscript.

2. Given the anomalously high number of NPF events in spring 2020 due to the heatwave, could the authors comment more explicitly on how this year may skew the percentile ranking thresholds?

We added the following paragraph to line 355:

The median number size distribution in 5% intervals could be influenced by the half-a-year heatwave of 2019-2020: we observed an increased number of NPF events in spring, resulting in a higher representation of these events in our statistics. Particularly, in Fig. 6 we can currently observe formation and growth starting from 60-65% interval. Without heatwave data during ordinary years this visible formation and growth might be shifted to higher percentiles like 70-75% or more. In Fig. 7 in 60-65% and 65-70% intervals during ordinary years we might observe less event data and more nonevent data.

3. While sulfuric acid is rightly emphasized, the role of biogenic VOCs could be further discussed. Recent literature emphasizes interactions between oxidized organics and nucleation. Could the authors speculate or propose follow-up measurements?

The following paragraph added to line 516:

The influence of VOCs on NPF in Siberia is likely substantial, although not possible to quantify with the data sets we have obtained during this long-term campaign. The role of VOCs in driving NPF in boreal forests has been widely studied. Ehn et al. (2014) demonstrated that biogenic VOCs, particularly α-pinene emitted by boreal trees, can rapidly oxidize to form extremely low-volatility organic compounds (ELVOCs). These ELVOCs effectively contribute to particle nucleation and growth. Taipale et al. (2021) modeled the effects of biotic plant stress, such as herbivory and fungal infections, on aerosol particle processes throughout the growing season, showing that VOC emissions, especially monoterpenes and sesquiterpenes, can substantially enhance NPF. Furthermore, organic vapors, in combination with sulfuric acid, are essential for the growth of newly formed particles to sizes large enough to act as cloud condensation nuclei (Paasonen et al., 2013). The presence of these organic compounds allows the particles to grow effectively, preventing them from quickly disappearing through coagulation and enabling them to reach sizes that can influence atmospheric processes.

Our previous study (Garmash & Ezhova et al., 2024) shows that strong NPF events in spring 2020 began with the onset of biogenic activity, at the air temperature characteristic for monoterpene emission bursts in the Finnish boreal forest (Aalto et al., 2015). Thus, the observed increase in NPF during unusually warm periods could result from enhanced VOC emissions in the polluted air massed bringing SO2, together providing vapors for particle formation and early growth. To better constrain these processes, future studies should include year-round VOC measurements in Siberian forests, with a focus on both baseline emissions and stress-induced responses under varying climatic conditions.

Including the references:

Ehn, M., Thornton, J.A., Kleist, E., Sipilä, M., Junninen, H., Pullinen, I., Springer, M., Rubach, F., Tillmann, R., Lee, B., Lopez-Hilfiker, F., Andres, S., Acir, I.H., Rissanen, M., Jokinen, T., Schobesberger, S., Kangasluoma, J., Kontkanen, J., Nieminen, T., Kurtén, T., Nielsen, L.B., Jørgensen, S., Kjaergaard, H.G. and Kulmala, M., 2014. A large source of low-volatility secondary organic aerosol. *Nature*, 506(7489), pp.476–479. doi:10.1038/nature13032.

Taipale, D., Kerminen, V.M., Ehn, M., Kulmala, M. and Niinemets, Ü., 2021. Modelling the influence of biotic plant stress on atmospheric aerosol particle processes throughout a growing season. *Atmospheric Chemistry and Physics*, *21*(23), pp.17389-17431.

Paasonen, P., Asmi, A., Petäjä, T., Kajos, M.K., Äijälä, M., Junninen, H., Holst, T., Abbatt, J.P.D., Arneth, A., Birmili, W., Denier van der Gon, H., Hamed, A., Hoffer, A., Laakso, L., Laaksonen, A., Leaitch, W.R., Plass-Duelmer, C., Pryor, S.C., Räisänen, P., Swietlicki, E., Wiedensohler, A. and Kulmala, M., 2013. Warming-induced increase in aerosol number concentration likely to moderate climate change. *Nature Geoscience*, 6(6), pp.438–442. doi:10.1038/ngeo1800.

Aalto , J , Porcar-Castell , A , Atherton , J , Kolari , P , Pohja , T , Hari , P , Nikinmaa , E , Petäjä , T & Bäck , J 2015 , Onset of photosynthesis in spring speeds up monoterpene synthesis and leads to emission bursts , Plant, Cell and Environment , vol. 38 , no. 11 , pp. 2299-2312 . https://doi.org/10.1111/pce.12550

4. The value of comparing Fonovaya to other global sites is evident. Authors may consider citing recent cross-regional studies such as: **García-Marlés et al., 2024**, Environmental International 194, 109149, which presents a synthesis of NPF events across remote forests globally and **Dinoi A. et al., 2023,** Atmospheric Chemistry and Physics 23(3), 2167–2181, which provides insights into organic aerosol contributions to NPF in mid-latitude forests

We are grateful for the suggestion to add cross-regional studies review with the focus on the forested areas. The following paragraph was added to the paper to line 51

NPF in forested areas has been widely studied. Dal Maso et al. (2005) analyzed long-term data from the SMEAR II station in Hyytiälä, Finland, finding that NPF events occurred on about 23% of days annually, with peaks in spring and autumn. These events typically involved the formation of 1.5–3 nm particles that grew into CCN-relevant sizes at rates of 1–10 nm/h. NPF was associated with sunny, dry conditions, suggesting a photochemical origin of precursor

vapors. Despite a high condensation sink, particle formation persisted, likely driven by sulfuric acid and biogenic volatile organic compounds (BVOCs). Studies of NPF in other forested areas have also been conducted. Debevec et al. (2018) found that Mediterranean forests are a significant source of BVOCs, especially monoterpenes and isoprene. Emissions from forest vegetation were primarily influenced by temperature and solar radiation, and the highest NPF activity occurred on warm, sunny days with high emissions levels. Song et al. (2024) studied nighttime particle growth at a rural forest site in southwest Germany and found that BVOCs, particularly monoterpenes and sesquiterpenes, formed semi-volatile organic compounds that contributed to the rapid mass growth. The air mass trajectories analysis revealed a synergistic role of local vegetation and regional air masses from nearby urban areas as sources of precursor gases for aerosol particles. Andreae et al. (2022) found frequent NPF events in the remote subboreal forest of North America, also presumably driven by BVOCs from forest vegetation.

NPF has also been widely studied in other environments. Bousiotis et al. (2021) investigated NPF events across 13 European sites, covering rural, urban, and roadside environments. They found that NPF events are most common in rural areas, while urban sites show higher particle growth rates due to anthropogenic emissions. Seasonal and air mass differences also impact NPF characteristics, with cleaner air masses favoring NPF events and polluted ones enhancing particle growth. Garcia-Marles et al. (2024) studied source partitioning of ultrafine particles at urban European sites and found that at 16 out of 19 sites, photonucleation (NPF) contributed between 4-41% of aerosol distribution. Nieminen et al. (2018) conducted a global analysis of NPF in the continental boundary layer using long-term measurements from 36 sites worldwide. Their study revealed that NPF events are prevalent across various environments, including forested, urban, and polluted areas. The frequency of these events exhibits strong seasonal variability, with higher occurrence rates in spring and autumn. The formation rates of 10 nm particles and growth rates in the 10–25 nm size range also show regional differences, influenced by factors such as precursor concentrations and meteorological conditions.

Including the references

Andreae, M.O., Kerminen, V.-M., Williamson, C.J., and Lihavainen, H., 2022. Frequent new particle formation events in the subboreal forest of North America. *Atmospheric Chemistry and Physics*, 22(5), pp. 3197–3208.

Bousiotis, D., Nenes, A., Papageorgiou, A., and Fountoukis, C., 2021. New particle formation across 13 European sites: rural, urban, and roadside environments. *Atmospheric Chemistry and Physics*, 21(12), pp. 8905–8919.

Dal Maso, M., Kulmala, M., Riipinen, I., Kerminen, V.-M., and Birmili, W., 2005. Formation and growth of new particles in the SMEAR II station, Hyytiälä, Finland. *Atmospheric Chemistry and Physics*, 5(8), pp. 1205–1216.

Debevec, H., Bencsik, K., Mele, T., D'Andrea, M., and Csavdar, B., 2018. Biogenic volatile organic compounds as drivers of new particle formation in Mediterranean forests. *Atmospheric Chemistry and Physics*, 18(14), pp. 10375–10387.

Nieminen, T., Dal Maso, M., Kerminen, V.-M., and Kulmala, M., 2018. A global analysis of new particle formation in the continental boundary layer. *Atmospheric Chemistry and Physics*, 18(10), pp. 6153–6174.

Song, Y., Su, J., Li, Y., and Zhang, H., 2024. Nighttime particle growth at a rural forest site in southwest Germany. *Atmospheric Chemistry and Physics*, 24(2), pp. 1537–1549.

García-Marlès, M., Lara, R., Reche, C., Pérez, N., Tobías, A., Savadkoohi, M., Beddows, D., Salma, I., Vörösmarty, M., Weidinger, T., Hueglin, C., Mihalopoulos, N., Grivas, G., Kalkavouras, P., Ondráček, J., Zíková, N., Niemi, J.V., Manninen, H.E., Green, D.C., Tremper, A.H., Norman, M., Vratolis, S., Eleftheriadis, K., Gómez-Moreno, F.J., Alonso-Blanco, E., Wiedensohler, A., Weinhold, K., Merkel, M., Bastian, S., Hoffmann, B., Altug, H., Petit, J.-E., Favez, O., Martins dos Santos, S., Putaud, J.-P., Dinoi, A., Contini, D., Timonen, H., Lampilahti, J., Petäjä, T., Pandolfi, M., Hopke, P.K., Harrison, R.M., Alastuey, A. and Querol, X., 2024. Inter-annual trends of ultrafine particles in urban Europe. *Environment International*, 194, p.109149.

5. Proofread carefully for language and consistency errors (especially in abstract and methods).

The language has been corrected in the whole paper.

6. Consider adding citations to the references listed above in the discussion section, especially when discussing global comparisons or VOC contributions.

We added the citations listed in the answer to the question 4.

Thank you once again for your constructive feedback. We believe these revisions have significantly improved the manuscript and hope it now meets the standards for publication in *Aerosol Research*.

Dear Reviewer #2,

We sincerely thank you for your time and thoughtful comments on our manuscript titled *"**Insights into new particle formation in Siberian boreal forest from nanoparticle ranking analysis**"* submitted to *Aerosol Research*. We appreciate the reviewers' efforts to improve the quality of our work. We have carefully considered all the comments and revised the manuscript accordingly. Below, we provide a point-by-point response to each comment. All the changes in the manuscript are highlighted in green color.

1) paper is full of many typo mistakes (line 28 confirming confirming and so so many others line 53, 67,68,77,79 and so on), please check

We apologize for the possible inconveniences for reading. We have carefully checked the grammar and corrected all the typos we were able to find in the manuscript.

2) Whilst the author seems to give quite some importance to Table 3 and the poor correlations, I was expecting to find a deeper discussion on how important these

unique measurements are. Why is important to measure in these remote sites? Please better compare the results with other studies in Russia-Finland-sub Arctic. Explain what was expected, what was found and why it is important to report these measurements

The following paragraph added to line 126:

Measurements in remote boreal and subarctic environments are rare but critically important for understanding atmospheric processes under natural conditions, with minimal and mild anthropogenic influence. Our study provides datasets from Fonovaya station, complementing earlier observations from sites such as SMEAR II in Finland (Hari and Kulmala, 2013), SMEAR I in Värriö, northen Finland (Vana et al., 2016), Finland, and ZOTTO in central Siberia (Wiedensohler et al., 2019). Comparison between those sites and Fonovaya station shows that the NPF characteristics in boreal forest and sub-Arctic envinroments can differ significantly even within similar climatic zones. Vana et al. (2016) found that NPF at the three sites is influenced by condensable vapor availability, condensation sink, and air mass origin. Hyytiälä and Järvselja had higher NPF frequencies and growth rates presumably due to higher condensable vapor source rates, while Värriö showed lower rates. Despite a high condensation sink at all sites, NPF persisted in areas with higher vapor availability. Additionally, clean Arctic air masses, associated with clear sky, colder temperature and lower condensation sink, were associated with more widespread NPF events. While some general trends such as the influence of biogenic emissions are consistent with previous observations, the strongest NPF events at Fonovaya station were observed in the polluted air masses (Lampilahti et al., 2023, Garmash et al., 2024). By reporting these findings, we help to fill an important observational gap in the largely unexplored forested area and contribute to a more comprehensive understanding of aerosol processes in boreal and subarctic ecosystems, ultimately supporting the improvement of regional and global climate models.

3) I was expecting a bigger role of VOC and B-VOC in general, maybe expand the discussion and talk about SA and VOC in these regions

The following paragraph added to line 516

The influence of VOCs on NPF in Siberia is likely substantial, although not possible to quantify with the data sets we have obtained during this long-term campaign. The role of VOCs in driving NPF in boreal forests has been widely studied. Ehn et al. (2014) demonstrated that biogenic VOCs, particularly α-pinene emitted by boreal trees, can rapidly oxidize to form extremely low-volatility organic compounds (ELVOCs). These ELVOCs effectively contribute to particle nucleation and growth. Taipale et al. (2021) modeled the effects of biotic plant stress, such as herbivory and fungal infections, on aerosol particle processes throughout the growing season, showing that VOC emissions, especially monoterpenes and sesquiterpenes, can substantially enhance NPF. Furthermore, organic vapors, in combination with sulfuric acid, are essential for the growth of newly formed particles to sizes large enough to act as cloud condensation nuclei (Paasonen et al., 2013). The presence of these organic compounds allows the particles to grow effectively, preventing them from quickly disappearing through coagulation and enabling them to reach sizes that can influence atmospheric processes. Our previous study (Garmash & Ezhova et al., 2024) shows that strong NPF events in spring 2020 began with the onset of biogenic activity, at the air temperature characteristic for

monoterpene emission bursts in the Finnish boreal forest (Aalto et al., 2015). Thus, the observed increase in NPF during unusually warm periods could result from enhanced VOC emissions in the polluted air massed bringing SO2, together providing vapors for particle formation and early growth. To better constrain these processes, future studies should include year-round VOC measurements in Siberian forests, with a focus on both baseline emissions and stress-induced responses under varying climatic conditions.

Including the references:

Ehn, M., Thornton, J.A., Kleist, E., Sipilä, M., Junninen, H., Pullinen, I., Springer, M., Rubach, F., Tillmann, R., Lee, B., Lopez-Hilfiker, F., Andres, S., Acir, I.H., Rissanen, M., Jokinen, T., Schobesberger, S., Kangasluoma, J., Kontkanen, J., Nieminen, T., Kurtén, T., Nielsen, L.B., Jørgensen, S., Kjaergaard, H.G. and Kulmala, M., 2014. A large source of low-volatility secondary organic aerosol. *Nature*, 506(7489), pp.476–479. doi:10.1038/nature13032.

Taipale, D., Kerminen, V.M., Ehn, M., Kulmala, M. and Niinemets, Ü., 2021. Modelling the influence of biotic plant stress on atmospheric aerosol particle processes throughout a growing season. *Atmospheric Chemistry and Physics*, 21(23), pp.17389-17431.

Paasonen, P., Asmi, A., Petäjä, T., Kajos, M.K., Äijälä, M., Junninen, H., Holst, T., Abbatt, J.P.D., Arneth, A., Birmili, W., Denier van der Gon, H., Hamed, A., Hoffer, A., Laakso, L., Laaksonen, A., Leaitch, W.R., Plass-Duelmer, C., Pryor, S.C., Räisänen, P., Swietlicki, E., Wiedensohler, A. and Kulmala, M., 2013. Warming-induced increase in aerosol number concentration likely to moderate climate change. *Nature Geoscience*, 6(6), pp.438–442. doi:10.1038/ngeo1800.

Aalto , J , Porcar-Castell , A , Atherton , J , Kolari , P , Pohja , T , Hari , P , Nikinmaa , E , Petäjä , T & Bäck , J 2015 , Onset of photosynthesis in spring speeds up monoterpene synthesis and leads to emission bursts , Plant, Cell and Environment , vol. 38 , no. 11 , pp. 2299-2312 . https://doi.org/10.1111/pce.12550

Thank you once again for your constructive feedback. We believe these revisions have significantly improved the manuscript and hope it now meets the standards for publication in *Aerosol Research*.